

# Alcohol consumption and sleep deprivation among Ghanaian adults: Ghana Demographic and Health Survey

Sanni Yaya[1], Ruoxi Wang[2], Tang Shangfeng[2] and Bishwajit Ghose[1]

[1] School of International Development and Global Studies, University of Ottawa, Ottawa, ON, Canada
[2] School of Medicine and Health Management, Huazhong University of Science and Technology, Wuhan, China

## ABSTRACT

**Abstract:** Heavy consumption of alcohol has shown to be associated with sleep disturbances among adult and elderly people in high income settings. So far, the relationship between alcohol drinking and sleeping pattern has not been studied in an African setting. Therefore, in this study we investigated whether alcohol consumption has any influence on sleeping hours among adult men and women in Ghana.

**Methods:** Data for this survey were extracted from Ghana Demographic and Health Survey (GDHS 2008). GDHS is the only cross-sectional survey conducted on men and women aged above 15 years that collected information on variables such as sleeping hours and alcohol consumption. The analysis was controlled for various demographic, socioeconomic, household level factors, and smoking.

**Results:** Prevalence of sleeping 1–3 h, 4–6 h, and >7 h was respectively 1.5% (1.2–2.0), 14.1% (12–16.5), and 84.4% (82.1–86.4), and that of alcohol use was 26.9% (23.4–30.6). In the multivariable regression analysis, compared with non-drinkers, those reported drinking had significantly lower odds of sleeping for at least 7 h. In the adjusted model, drinkers had 0.8 times (adjusted OR = 0.803, (95% CI [0.690–0.935])) lower odds of sleeping for at least 7 h. The odds for sleeping 4–6 h were not statistically significant. In the stratified analysis, the odds of sleeping for at least seven were comparatively lower among women (adjusted OR = 0.657, (95% CI [0.509–0.849]) then among men (adjusted OR = 0.867, (95% CI [0.740–0.965]).

**Conclusion:** Men and women who reported consuming alcohol had significantly lower odds of getting adequate sleep (>7 h). The sleep-disrupting effect of alcohol appeared to be more prominent among women than among men. Currently there is not sufficient evidence on alcohol consumption and sleep disorder among Ghanaian population or any other country in the region. Further studies are required to understand sleeping patterns and the burden of alcohol drinking in this population to design intervention programs.

Corresponding author
Bishwajit Ghose,
brammaputram@gmail.com

## INTRODUCTION

Alcohol consumption accounts for approximately 4% of all-cause global mortality and 5% of the global disease burden (*Beaglehole & Bonita, 2009*). As such, alcohol is recognised by healthcare systems as a major risk factor for a host of physical and psychological illnesses, injuries, and self-harm, as well as social issues that create significant bearing on indicators such as potential years of life lost, and disability-adjusted life years (*Lam & Chim, 2010*; *Collin, Hill & Smith, 2015*; *Rabiee et al., 2017*; *Shield et al., 2012*). Advances in molecular pathology has greatly facilitated the understanding of the underlying mechanisms through which alcohol affects major regulatory systems leading to endocrine disorder and associated consequences such as disruption of the circadian rhythm (Process C) (*Emanuele & Emanuele, 1997*; *De Coster & Van Larebeke, 2012*; *Rachdaoui & Sarkar, 2013*, *2017*). As an endocrine-disrupting agent, and complex interaction with body's circadian rhythm, alcohol has become a subject of widespread interest among health scientists (*Wasielewski & Holloway, 2001*).

One of the key issues concerned with the alcohol-induced disruption of the circadian rhythm (clock–hormone interaction) is altered sleep architecture (e.g. nocturnal sleep disruption, excessive daytime sleepiness) (*Jafari Roodbandi, Choobineh & Daneshvar, 2015*) as it plays a key role in the regulation of sleep homeostasis, disruption of which is a primary cause of sleeplessness (*Thakkar, Sharma & Sahota, 2015*). General sleep deprivation is a growing health concern and despite that it remains largely ignored in the population health promotion agenda (*Moloney, Konrad & Zimmer, 2011*). The cumulative effects of sleep disorders have been shown to trigger a wide-ranging physical and mental health consequences, poor academic and workplace productivity, decreased quality-of-life and well-being, and high healthcare expenditures (*Soares, 2005*; *Hillman, Murphy & Pezzullo, 2006*; *Johansson et al., 2010*; *Ishak et al., 2012*). For instance, the overall cost of sleep disorders in Australia was dollar estimated to be $7494 million (*Hillman, Murphy & Pezzullo, 2006*), while in the USA, it was estimated that societal costs of alcohol-related sleep disorders exceeds $18 billion (*Thakkar, Sharma & Sahota, 2015*).

The pathophysiology of sleep disorder is multifactorial, and the risk factors vary among people across different cultures, demographics, environment, socioeconomic situation. Apart from that, a growing body of research suggests that the epidemic of sleep disorder is being fuelled by changing lifestyle behaviour, for example, dietary habits, drug abuse, alcohol consumption, and smoking (*Mahfoud et al., 2009*; *Krishnan, Dixon-Williams & Thornton, 2014*; *Pot, 2017*). As a result, pharmacological interventions of sleep disorder are becoming increasingly popular, however, with multitude of potential disadvantages. In addition, various behavioural intervention techniques are also being tested such as sleep restriction therapy, sleep hygiene, paradoxical intention therapy, cognitive behavioural therapies (*Sharma & Andrade, 2012*). Despite these advances, treating alcohol-induced sleep problems remains a challenging one, particularly due to the complexities during withdrawal (*Brower, 2001*).

Despite the emerging evidences on the adverse impacts of alcohol, the rate of consumption is increasing in low-and-middle-income-countries (LMICs) (*Martinez et al., 2011*;

*Tian & Liu, 2011*). According to World Health Organization estimates, Sub-Saharan Africa ranks among the regions with highest per capita consumption of alcohol in the world (*WHO, 2013*). Data on sleep health, on the other hand, is extremely scarce for Ghana. However, it is assumable that the population is experiencing an increasing prevalence of sleep disorder given the growing popularity of alcoholic beverages. To address the current research gap, we undertook this study based on publicly available data from Ghana Demographic and Health Survey (GDHS) conducted in 2008. Although more recent survey data have been published, information on sleeping hours were included only in the 2008 survey. The main objectives of the study were to investigate the prevalence of sleep deprivation, as well its association with alcohol consumption status among men and women.

## METHODS

### Survey methods

The 2008 GDHS was the fifth to be undertaken in the country. This was a country-wide survey that covered all 10 regions in the country. The main components of DHS include maternal and child health indicators such as fertility, family planning, breastfeeding, nutritional status of women and young children, childhood mortality. The survey was carried out by the Ghana Statistical Service (GSS) and the Ghana Health Service (GHS) with technical support from MEASURE DHS programme and financial assistance from the United States Agency for International Development. The survey employed a two-stage sampling design to ensure representativeness of the population. The first stage involved selecting clusters (in total 412 were selected) from a master sampling frame constructed from the 2000 Ghana Population and Housing Census. In the second stage, a systematic selection of households (30 households from each cluster) was conducted from each cluster. A total of 5,096 women and 4,769 men were identified as eligible for survey, of who interviews were completed with 4,916 women (response rate of 97%) and 4,568 men (response rate of 96%). Field work for the survey lasted from 8 September to 25 November 2008. Further details about the survey are available at: GSS, GHS, and Inner city fund (ICF) Macro. 2009. GDHS 2008. Accra, Ghana: GSS, GHS, and ICF Macro.

### Variable used in the study

Outcome variable was self-reported hours of sleeping. This was assessed by asking the participants: *How many hours do you rest a day, including naps and sleep both during day and night?* Responses were collected in the following categories: 1–3 h, 4–6 h, 7–9 h, 10 and don't know. For the purpose of this study the categories were merged followingly: 1–3 h, 4–6 h, 6 + h. This was guided by the recommendations by WHO on sleep hours which maintains that adults should sleep seven or more hours per night on a regular basis to promote optimal health (*Watson et al., 2015*). Sleeping less than 7 h per night has been shown to be associated with various health consequences including obesity, impaired immune function, diabetes, hypertension, heart disease and stroke, depression, and increased mortality rates (*Watson et al., 2015*).

Main explanatory variable was alcohol drinking. Participants were asked whether or not they drink alcoholic beverages. As the information on the exact volume of consumption was not available, the answers were categorized as: Drinker if responded as yes, and Non-drinker if no.

To measure the independent association between alcohol drinking and sleeping hours, the analysis was adjusted for a range of potentially confounding variables such demographic, household, and socioeconomic status. For the purpose of selecting confounding variables that were available on the dataset, we conducted a literature review to identify which ones are theoretically related with alcohol drinking and sleep disturbance. Based on the review the following variables were finally selected for the analysis: Age: 15–19/20–24/25–29/30–34/35–39/40–44/44+; Sex: Male/Female; Residency: urban/rural; Religion: Christian/Non-Christian; Ethnicity: Akan/Ewe/Mole-Dagbani/Other; Education: No education/primary/Secondary/Higher; No. of household member: 4/6/8/8+; Has electricity: No/Yes; No. of rooms for sleeping: 2/4/6/6+; Has Bed: No/Yes; Has bed net: No/Yes; Wealth index: Poorest/Poorer/Middle/Richer/Richest; Working: No/Yes; Smokes cigar: No/Yes.

## Data analysis

Data analyses were performed with SPSS version 24 for windows. The datasets (men's and women's dataset) were checked for outliers, multicollinearity among variables, and then merged to perform pooled analysis. As the survey used multistage sampling techniques, regular analytical procedures were not suitable as they cannot adjust for cluster effects. In order to adjust for this effect, the dataset was prepared for complex analysis accounting for the primary sampling units, sample strata, and weight. This allowed complex sample analysis which is recommended for DHS data. After preparing the file, descriptive analyses were carried out to calculate the basic sociodemographic characteristics of the sample population and the prevalence of sleeping hours. These results were presented as percentages with 95% CIs. Following that, Chi-square bivariate tests were performed to calculate the prevalence of sleeping hours across the explanatory variables and the significance of these associations. Variables significant at $p < 0.25$ were entered into the multivariate model. We used multinomial logistic regression analyses to calculate the adjusted odds ratios of the association between the alcohol consumption and sleeping hours while controlling for the potentially confounding variables. Four different models were formed to see the contribution of the variables at certain types on the odds ratios. The level of significance was set at $p < 0.05$ for regression analyses.

## Ethics statement

DHS surveys are approved by ICF international, USA. All participants gave informed consent before taking part in the survey. Data are available in the public domain in anonymised form, therefore no additional approval was necessary.

# RESULTS

## Descriptive statistics

Basic sociodemographic profile of the participants was presented in Table 1. In total 4,546 men and 4,907 women were included in the analysis. Mean age of the participants

**Table 1 Sample characteristics. GDHS 2008.**

|  | N = 9,453 | % |
|---|---|---|
| **Age** | | |
| 15–19 | 1,974 | 20.9 |
| 20–24 | 1,571 | 16.6 |
| 25–29 | 1,422 | 15.0 |
| 30–34 | 1,153 | 12.2 |
| 35–39 | 1,145 | 12.1 |
| 40–44 | 872 | 9.2 |
| 45 or above | 1,316 | 13.9 |
| **Sex** | | |
| Male | 4,546 | 48.1 |
| Female | 4,907 | 51.9 |
| **Residency** | | |
| Urban | 4,066 | 43.0 |
| Rural | 5,387 | 57.0 |
| **Religion** | | |
| Christian | 6,714 | 71.0 |
| Non-Christian | 2,739 | 29.0 |
| **Ethnicity** | | |
| Akan | 3,962 | 41.9 |
| Ewe | 1,288 | 13.6 |
| Mole-Dagbani | 2,158 | 22.8 |
| Other | 2,045 | 21.6 |
| **Education** | | |
| No education | 2,021 | 21.4 |
| Primary | 1,723 | 18.2 |
| Secondary | 5,155 | 54.5 |
| Higher | 554 | 5.9 |
| **No. of household member** | | |
| 4 | 4,426 | 46.8 |
| 6 | 2,608 | 27.6 |
| 8 | 1,446 | 15.3 |
| 8+ | 973 | 10.3 |
| **Has electricity** | | |
| No | 4,215 | 44.6 |
| Yes | 5,235 | 55.4 |
| **Rooms used for sleeping** | | |
| 2 | 6,878 | 72.8 |
| 4 | 1,850 | 19.6 |
| 6 | 390 | 4.1 |
| 6+ | 335 | 3.5 |

(Continued)

| Table 1 (continued). | | |
| --- | --- | --- |
| | N = 9,453 | % |
| **Has bed** | | |
| No | 1,481 | 15.7 |
| Yes | 7,972 | 84.3 |
| **Has bed net** | | |
| No | 4,193 | 44.4 |
| Yes | 5,260 | 55.6 |
| **Wealth index** | | |
| Poorest | 2,167 | 22.9 |
| Poorer | 1,785 | 18.9 |
| Middle | 1,628 | 17.2 |
| Richer | 1,979 | 20.9 |
| Richest | 1,894 | 20.0 |
| **Working** | | |
| No | 2,101 | 22.2 |
| Yes | 7,352 | 77.8 |
| **Smokes cigar** | | |
| No | 9,059 | 95.8 |
| Yes | 394 | 4.2 |
| **Drinks alcohol** | | |
| No | 6,809 | 72.0 |
| Yes | 2,644 | 27.9 |

were 28.99 years (SD 9.7), 31.67 years (SD 12.24) among men and 28.99 (SD 9.70) among women.

Most of the participants were in the age groups of 15–19 years, were female, rural residents, followers of Christianity, of Akan ethnicity, had secondary level education. Most of the households had four members, had access to electricity, had two living rooms, possessed bed, bed net, from richer to richest households. Majority of the participants had employment, were non-smoker and non-drinker.

## Sleeping hours

More than four-fifth of the participants (84.4%, (95% CI [82.1–86.4])) reported sleeping for more than 7 h on an average day. Prevalence of sleeping 1–3 h, 4–6 h, and ≥7 h was respectively 1.5% (95% CI [1.2–2.0]) and 14.1% (95% CI [12–16.5]). From Table 2 it appears that the prevalence of sleeping ≥7 h were comparatively higher among those aged below 30 years, female, rural residents, non-Christian, of Akan ethnicity, had secondary level education, lived in households composed of four members, had access to electricity, had two rooms for sleeping, possessed bed and bed net, non-poor households (richest/richer), had employment, were non-smoker and non-drinker.

### Association between alcohol drinking and adequacy of sleeping hours

Results of multivariable regression analysis on the association between alcohol drinking of any frequency and sleeping hours were presented in Table 3. In the pooled analysis,

**Table 2** Sleeping hours across the explanatory variables. GDHS 2008.

| Variables | Sleeping hours | | | p-value |
|---|---|---|---|---|
| | 1–3<br>1.5% (1.2–2.0) | 4–6<br>14.1% (12–16.5) | >7<br>84.4% (82.1–86.4) | |
| **Age** | | | | |
| 15–19 | 19.5 (13.5–27.5) | 12.7 (10.1–15.7) | 21.7 (20.6–23.0) | <0.001 |
| 20–24 | 11.8 (6.7–19.8) | 14.1 (12.0–16.4) | 17.2 (15.9–18.5) | |
| 25–29 | 18.0 (12.0–26.1) | 14.2 (12.0–16.7) | 15.5 (14.6–16.5) | |
| 30–34 | 11.9 (7.4–18.6) | 15.8 (13.3–18.7) | 11.8 (11.0–12.7) | |
| 35–39 | 12.3 (7.3–19.8) | 15.2 (13.4–17.3) | 11.8 (11.0–12.6) | |
| 40–44 | 7.5 (4.0–13.8) | 10.6 (8.9–12.6) | 8.9 (8.2–9.6) | |
| 45–49 | 19.0 (12.7–27.5) | 17.4 (15.4–19.6) | 13.0 (11.5–14.8) | |
| **Sex** | | | | |
| Male | 45.8 (29.3–63.2) | 57.9 (41.7–72.5) | 46.5 (33.7–59.7) | <0.001 |
| Female | 54.2 (36.8–70.7) | 42.1 (27.5–58.3) | 53.5 (40.3–66.3) | |
| **Residency** | | | | |
| Urban | 41.6 (27.3–57.4) | 54.3 (37.0–70.7) | 46.5 (35.3–58.1) | <0.001 |
| Rural | 58.4 (42.6–72.7) | 45.7 (29.3–63.0) | 53.5 (41.9–64.7) | |
| **Religion** | | | | |
| Christian | 55.7 (44.4–66.5) | 24.6 (21.1–28.4) | 24.6 (20.0–29.9) | <0.001 |
| Non-Christian | 44.3 (33.5–55.6) | 75.4 (71.6–78.9) | 75.4 (70.1–80.0) | |
| **Ethnicity** | | | | |
| Akan | 38.0 (26.6–50.9) | 48.1 (40.3–56.1) | 49.5 (42.6–56.4) | <0.001 |
| Ewe | 7.8 (4.0–14.6) | 18.0 (12.7–24.9) | 13.2 (9.3–18.3) | |
| Mole-Dagbani | 35.8 (25.3–47.9) | 11.1 (8.3–14.7) | 16.9 (12.4–22.6) | |
| Other | 18.4 (10.6–30.0) | 22.8 (19.1–26.9) | 20.4 (16.8–24.5) | |
| **Education** | | | | |
| No education | 30.9 (22.2–41.3) | 17.4 (13.1–22.7) | 17.5 (13.6–22.2) | <0.001 |
| Primary | 18.0 (12.5–25.1) | 14.6 (11.5–18.3) | 17.9 (16.0–19.9) | |
| Secondary | 45.2 (34.5–56.3) | 57.3 (52.9–61.6) | 59.0 (54.3–63.5) | |
| Higher | 5.9 (2.9–11.6) | 10.7 (6.4–17.3) | 5.6 (4.5–7.1) | |
| **No. of household member** | | | | |
| 4 | 40.9 (32.9–49.5) | 53.9 (48.7–59.0) | 48.9 (45.5–52.4) | <0.001 |
| 6 | 34.4 (25.3–44.6) | 28.1 (24.7–31.7) | 27.4 (25.7–29.1) | |
| 8 | 9.9 (6.0–16.0) | 10.9 (8.8–13.4) | 14.9 (13.3–16.7) | |
| 8+ | 14.8 (8.5–24.7) | 7.1 (5.6–9.0) | 8.8 (7.0–10.9) | |
| **Has electricity** | | | | |
| No | 53.4 (39.6–64.7) | 33.0 (22.9–40.9) | 39.5 (32.9–46.5) | <0.001 |
| Yes | 46.6 (34.1–59.4) | 67.0 (57.0–75.7) | 59.5 (52.3–66.0) | |
| **Rooms used for sleeping** | | | | |
| 2 | 61.5 (49.7–72.2) | 77.0 (73.1–80.5) | 75.9 (72.4–79.1) | <0.001 |
| 4 | 20.2 (12.4–31.1) | 17.2 (14.1–20.8) | 17.7 (15.5–20.1) | |
| 6 | 8.8 (5.0–14.9) | 2.8 (2.0–3.8) | 3.4 (2.7–4.3) | |
| 6+ | 9.5 (5.1–17.0) | 3.0 (2.1–4.3) | 3.0 (2.3–3.9) | |

(Continued)

## Table 2 (continued).

| Variables | Sleeping hours | | | p-value |
|---|---|---|---|---|
| | 1–3<br>1.5% (1.2–2.0) | 4–6<br>14.1% (12–16.5) | >7<br>84.4% (82.1–86.4) | |
| **Has bed** | | | | |
| No | 12.8 (6.8–22.8) | 8.9 (6.5–12.2) | 12.7 (9.6–16.4) | <0.001 |
| Yes | 87.2 (77.2–93.2) | 91.1 (87.8–93.5) | 87.3 (83.6–90.4) | |
| **Has bed net** | | | | |
| No | 45.8 (36.0–55.9) | 48.5 (43.7–53.3) | 47.2 (43.3–51.1) | 0.372 |
| Yes | 54.2 (44.1–64.0) | 51.5 (46.7–56.3) | 52.8 (48.9–56.7) | |
| **Wealth index** | | | | |
| Poorest | 31.6 (20.1–45.9) | 13.0 (8.5–19.3) | 17.1 (12.1–23.6) | <0.001 |
| Poorer | 21.9 (14.3–32.1) | 15.1 (10.3–21.7) | 18.5 (15.3–22.3) | |
| Middle | 13.6 (8.4–21.2) | 17.7 (12.9–23.8) | 18.8 (16.0–22.0) | |
| Richer | 19.3 (11.8–30.0) | 24.0 (20.7–27.7) | 23.1 (20.0–26.6) | |
| Richest | 13.6 (7.9–22.4) | 30.2 (19.3–43.8) | 22.5 (16.4–30.0) | |
| **Working** | | | | |
| No | 25.3 (18.2–33.9) | 12.6 (10.9–14.6) | 23.9 (22.3–25.6) | <0.001 |
| Yes | 74.7 (66.1–81.8) | 87.4 (85.4–89.1) | 76.1 (74.4–77.7) | |
| **Smokes cigar** | | | | |
| No | 96.9 (91.8–98.9) | 95.9 (93.8–97.3) | 96.4 (95.0–97.5) | <0.001 |
| Yes | 3.1 (1.1–8.2) | 4.1 (2.7–6.2) | 3.6 (2.5–5.0) | |
| **Drinks alcohol** | | | | |
| No | 78.8 (69.8–85.6) | 66.5 (61.7–70.9) | 74.2 (70.5–77.4) | 0.124 |
| Yes | 21.2 (14.4–30.2) | 33.5 (29.1–38.3) | 25.8 (22.5–29.4) | |

## Table 3 Odds ratios of sleeping 4–6 h and 1–3 h among drinkers. GDHS 2008.

| | Model 1 | | Model 2 | | Model 3 | | Model 4 | |
|---|---|---|---|---|---|---|---|---|
| | ≥7 h | 4–6 h | ≥7 h | 4–6 h | ≥7 h | 4–6 h | ≥7 h | 4–6 h |
| **Overall** | | | | | | | | |
| Alcohol (No) | | | | | | | | |
| Yes | 0.692<br>(0.595–0.805) | 1.294<br>(0.836–2.004) | 0.712<br>(0.619–0.819) | 1.159<br>(0.726–1.849) | 0.729<br>(0.638–0.833) | 1.208<br>(0.751–1.945) | 0.803<br>(0.690–0.935) | 1.154<br>(0.713–1.868) |
| **Men** | | | | | | | | |
| Alcohol (No) | | | | | | | | |
| Yes | 0.820<br>(0.679–0.990) | 0.983<br>(0.554–1.747) | 0.847<br>(0.725–0.989) | 0.993<br>(0.543–1.813) | 0.846<br>(0.727–0.983) | 0.994<br>(0.545–1.815) | 0.867<br>(0.740–0.965) | 0.986<br>(0.529–1.840) |
| **Women** | | | | | | | | |
| Alcohol (No) | | | | | | | | |
| Yes | 0.664<br>(0.514–0.859) | 2.236<br>(0.867–5.772) | 0.640<br>(0.496–0.826) | 2.113<br>(0.816–5.469) | 0.653<br>(0.507–0.842) | 2.201<br>(0.855–5.668) | 0.657<br>(0.509–0.849) | 2.170<br>(0.840–5.606) |

Notes:
Regression analysis.
N.B. Reference category = 1–3 h. Model 1 = Age, sex; Model 2 = Model 1, residence, religion, ethnicity; Model 3 = Model 2; education, wealth status, employment; Model 4 = Model 3, no. of household members, has electricity, no. of rooms used for sleeping, has bed, smoking.

compared with non-drinking participants, those who drink had significantly lower odds of sleeping for at least 7 h. For example, in the fully adjusted model drinkers had 0.8 times (AOR = 0.803, (95% CI [0.690–0.935])) lower odds of sleeping for at least 7 h and 1.15 times (AOR = 1.154, (95% CI [0.713–1.868])) higher odds of sleeping for 4–6 h. However, the odds for sleeping 4–6 h were not statistically significant. In the stratified analysis, the strength of the association appeared to be higher among women compared with men. After adjusting for all the potential confounders, the odds of at least 7 h of sleeping was 0.65 times (AOR = 0.657, (95% CI [0.509–0.849])) lower compared with odds ratios of 0.86 (AOR = 0.867, (95% CI [0.740–0.965])) among men.

## DISCUSSION

### Main findings

In the present study we found more than a quarter of the participants reported ever drinking alcohol. Regarding sleep disturbance, more than one in 10 reported sleeping for less than 7 h, and a smaller fraction (1.5%) less than 6 h a day. Significant sociodemographic patterns were observed in the prevalence of adequate sleeping. Adolescents (15–19 years) had the highest rate of reporting adequate sleeping, followed by those in the age groups of 20–24 and 24–29 years. This percentage decreased until the age group of 45–49 years. Women had higher prevalence of sleeping more than seven and less than 4 h, while men had higher prevalence of sleeping 4–6 h. Geographic and ethnic differences were also prominent, rural residents had higher prevalence of sleeping for at least 7 h than urban residents. We also found significant association with socioeconomic factors such as education and working status, participants who had secondary level education had employment had higher percentages of sleeping for more than 7 h. Regarding household characteristics, living in households composed of less than five members, had access to electricity, had two rooms for sleeping, possessed bed, of higher wealth status, appeared to have positive association with adequate sleeping hours.

Our findings also indicated a positive association between alcohol intake and poor-quality sleep measured in terms of sleeping hours. Data on sleeping hours were collected as brackets (e.g. 1–3 h), and not as continuous measures, which prevented us from conducting a linear regression analysis which could have provided a more robust picture of the association between alcohol intake and sleep disturbance. One interesting aspect regarding the association is that alcohol use decreased the odds of sleeping for ≥7 only, not for 4–6 h. Of note, these variations persisted for all levels of adjustment (partial and full) as adding the variables in groups by types did not make any noticeable different in the effect size. Further studies will be required to explore the mechanisms of these variations.

### Previous findings and research directions

Epidemiological studies on nationally representative sample in the areas of sleep medicine is rare for sub-Saharan countries. In recent years the number of studies on sleep health on sub-national population has increased in Nigeria and South Africa, however, the studies vary in methodological approaches for measuring sleep quality and sleeping

hours. Therefore, the findings on the prevalence of adequate sleeping hours are hard to compare with previous researches. One Nigerian study conducted on school-going adolescents in Kano state reported that both the quality and quantity of sleep were suboptimal, and recommended more research attention on the topic (*Peter et al., 2017*). Similar findings was reported among on school-going adolescents in Ibadan (*Balogun, Alohan & Orimadegun, 2017*). Another study on university students found that mean duration of night sleep was 6.2 h (*Oluwole, 2010*). Regarding the association between alcohol intake and sleep disturbance, the findings of our study are comparable with those from several review (*Roehrs & Roth, 2001*) and original studies (*Stein & Friedmann, 2005*; *Singleton & Wolfson, 2009*; *Dangour et al., 2013*; *Kenney et al., 2014*; *Van Schrojenstein Lantman et al., 2017*). These studies are based on sample mostly from Asian and European population and involve social drinking instead of problem drinking. Synthesis of current literature reveals that problem alcohol drinking/alcohol abuse is more commonly associated with sleep disturbance compared with social drinking (*Brower, 2001*; *Arnedt, Conroy & Brower, 2007*; *Popovici & French, 2013*). Therefore, future studies investigating sleep quality/disorder in association with alcohol use should take into consideration the frequency and degree of drinking.

## General discussion

Worldwide, sleep deprivation is a growing public health concern among the scientific community owing to the long-terms consequences on health and quality of life at individual level, and economic costs in terms of productivity loss at national level. Although most of the findings originate from researches in high-income settings, the epidemic is not unique to the wealthy countries as poverty of sleep is being recognised to be a widespread issue in LMICs as well (*Shah, Bang & Bhagat, 2010*; *Kang et al., 2012*; *Stranges et al., 2012*). In Africa, on the other hand, the issue has remained underemphasised to date and is potentially contributing to the worsening health outcomes and unmet health need of the population. Therefore, it is recommended that universities and health research institutions conduct surveys on sleep health especially in the context of alcohol consumption and inform authorities for taking appropriate policy action (*Hahn, Woolf-King & Muyindike, 2011*; *O'Connell et al., 2013*; *Ferreira-Borges, Parry & Babor, 2017*).

## Strengths and limitations

This is the first study to explore the association between alcohol consumption and adequacy of sleeping hours in an African setting. Some of the strengths include the large sample size and inclusion of both men and women. We were able to control for a large number of potentially confounding variables such as demographic, socioeconomic, and household. Besides the strengths, there are several important limitations of this study. There was no information on the use of psychoactive medication or any other drug that could influence quality of sleeping. Self-reported measure of sleeping hours and alcohol intake are subject to recall error and reporting bias. Alcohol intake was categorised as 'drinker' and 'non-drinker,' which prevents making any concrete conclusion such as the minimum frequency and volume of drinking to affect sleeping hours. As the data were

cross-sectional, it does not guarantee any causality or directionality of the association between the explanatory and outcome variables. It is possible that people who were not able to maintain healthy sleeping habits were more prone to developing drinking behaviour. Another important limitation is that we could not adjust the analysis for factors such as the occurrence of other diseases that are known to affect sleep quality. The survey was conducted in 2008, therefore the prevalence rates may not represent the current scenario. Future studies should focus on collecting more precise information regarding drinking behaviour among men and women including the elderly population and investigate people's perception of the impact of drinking on subjective sleep quality.

## CONCLUSIONS

Our findings indicate that nearly one-fifth of adult men and women in Ghana are not getting adequate sleep. Sleep-deprivation is a multifactorial issue and can arise from various environmental, biological, and lifestyle related behaviours such as alcohol drinking. We also found a significantly positive association between alcohol consumption and inadequate sleeping hours which supports the previous findings that regular use of alcohol may lead to sleep difficulties in the long run. We further observed that the sleep-disrupting effect of alcohol was more prominent among women than among men. Further studies are required to explore the causes of varying sleeping patterns among men and women in relation to alcohol drinking. At the health policy making level, programs aimed at reducing alcohol consumption at national level may have positive outcomes on sleep health in the population.

## ACKNOWLEDGEMENTS

Authors acknowledge the generous provision of the data that has made the study possible.

### Funding
The authors received no funding for this work.

### Competing Interests
The authors declare that they have no competing interests.

### Author Contributions
- Sanni Yaya analysed the data, authored or reviewed drafts of the paper, approved the final draft.
- Ruoxi Wang contributed reagents/materials/analysis tools, authored or reviewed drafts of the paper, approved the final draft.
- Tang Shangfeng analysed the data, contributed reagents/materials/analysis tools, authored or reviewed drafts of the paper, approved the final draft.
- Bishwajit Ghose conceived and designed the experiments, performed the experiments, analysed the data, contributed reagents/materials/analysis tools, prepared figures and/or tables, authored or reviewed drafts of the paper, approved the final draft.
## Human Ethics

The following information was supplied relating to ethical approvals (i.e., approving body and any reference numbers):

DHS surveys are approved by ICF international, USA. All participants gave informed consent before taking part in the survey. Data are available in the public domain in anonymised form, therefore no additional approval was necessary.

## Data Availability

DHS Program: https://dhsprogram.com/what-we-do/survey/survey-display-301.cfm.

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
