# Peer review of "Alcohol consumption and sleep deprivation among Ghanaian adults: Ghana Demographic and Health Survey"

_PeerJ, doi:10.7717/peerj.5750_

## Round 0.1 · original submission · Minor Revisions

Thank you for your submission. Your paper requires some revisions to your introduction / review of the literature, and careful proof reading throughout. We look forward to receiving a revised version in due course.

·

Basic reporting

1. The introduction needs more detail. I suggest you develop lines 243-257 further for the introduction by reviewing sleep and alcohol studies conducted in Sub Saharan Africa countries.

2. Line 72-80: You provide information from a wide range of contexts. I suggest that you extrapolate on the effect and costs of alcohol related sleep disorders in Ghana as well.

3. There appears to be a contradiction in line 95-96 compared with your description of sleep studies in these regions in lines 243-252

4. Line 95-96, 282- 284. This assertion might be accurate for Ghana but not ‘in an African setting’. Several studies have explored alcohol and sleep in sub Saharan Africa for example in Nigeria. Consider conducting a literature review beyond Ghana, or restrict the argument in this section to Ghana.

5. Consider your use of acronyms by spelling them out at first use.

6. The English language should be improved. Consider rephrasing lines, 30, 72, 112, 172, 151, 200, 259, 261, 267, 294, 295.

7. Consider making available a link to the public DHS data.

Experimental design

'No comment'

Validity of the findings

Limit your discussion in line 270-272, to those supported by the results. Provide a link to the original DHS data.

Additional comments

This is a preliminary study in the right direction. I commend the authors for identifying and acknowledging the various limitations in the study and providing direction for further studies.

Reviewer 2 ·

Basic reporting

The English language should be improved to ensure that an
international audience can clearly understand your text.
Some examples where the language could be improved
include the abstract :In this study investigated whether current
alcohol consumption has any influence on sleeping hours among adult men and women in
Ghana" – the current phrasing makes
comprehension difficult.

Some citations were wrongly attributed, e.g. According to World Health
94 Organization estimates, Sub Saharan Africa (SSA) ranks among the regions with highest per capita consumption of alcohol in the world(Onwuka et al., 2016).
It would make sense to cite the original WHO document where this information can be found.

Also reference no 1. was wrongly stated. My PUBMED Search revealed that the correct citation is: Barrett PR, Horne JA, Reyner LA.Alcohol continues to affect sleepiness related driving impairment, when breath alcohol levels have fallen to near-zero.Hum Psychopharmacol. 2004 Aug;19(6):421-3.

Also reference no. 2 is supposed to be:

Wasielewski JA and Holloway, FA. Alcohol’s Interactions With Circadian Rhythms: A Focus on Body Temperature. Bethesda, MD, United States: National Institute of Alcohol Abuse and Alcoholism.

Experimental design

Original primary research within Aims and Scope of the journal.
Research question well defined, relevant & meaningful. It is stated how research fills an identified knowledge gap

The research question is not well defined. Please state the research question.

Validity of the findings

Data is robust, statistically sound, & controlled.

Additional comments

Your references do not meet the journal format. Also some of your references lack basic information even when they have been updated in PUBMED. Please go through all of your references and format very well. Your manuscript would required the assistance of a native English speaker to improve the language structure.

---

## Round 0.2 · accepted · Accept

Thank you for your re-submitted manuscript. You have clearly addressed the minor concerns of both reviewers, There are a couple of typographical errors, however these may be addressed at the proof stage.